# Experiences of oral pre-exposure prophylaxis (PrEP) use disclosure among South African adolescent girls and young women and its perceived impact on adherence

Danielle Giovenco[1,2]*, Katherine Gill[2], Lauren Fynn[2], Menna Duyver[2], Shannon O'Rourke[3], Ariane van der Straten[3,4], Jennifer F. Morton[5], Connie L. Celum[5,6], Linda-Gail Bekker[2]

1 Department of Epidemiology, University of North Carolina at Chapel Hill, Chapel Hill, NC, United States of America, 2 The Desmond Tutu HIV Centre, University of Cape Town, Cape Town, South Africa, 3 RTI International, Women's Global Health Imperative, Berkeley, CA, United States of America, 4 Center for AIDS Prevention Studies, Department of Medicine, UCSF, San Francisco, CA, United States of America, 5 Department of Global Health, University of Washington, Seattle, WA, United States of America, 6 Departments of Medicine and Epidemiology, University of Washington, Seattle, WA, United States of America

* dgiovenco@unc.edu

**Data Availability Statement:** Data from this study may contain identifying or sensitive patient

## Abstract

### Introduction

There is limited understanding of how social dynamics impact pre-exposure prophylaxis (PrEP) adherence among adolescent girls and young women (AGYW) in generalized HIV-epidemic settings. We examined experiences of oral PrEP use disclosure to various social groups with the goal of identifying supportive relationships that can be leveraged to promote adherence.

### Methods

We used qualitative methods to explore experiences disclosing PrEP use and the perceived impact of disclosure on adherence among 22 South African AGYW (16–25 years) taking daily oral PrEP. Serial in-depth-interviews (IDIs) were conducted 1-, 3-, and 12-months post-PrEP initiation. Respondents also self-reported their disclosures separately for various social groups and adherence was assessed using intracellular tenofovir-diphosphate levels.

### Results

Qualitative respondents had a median age of 20.5 years and reported disclosing their PrEP use to friends (n = 36 total disclosures), partners, siblings, other family members (n = 24 disclosures each), and parents (n = 19 disclosures). IDI data revealed that parents and partners provided the most support to respondents and a lack of support from these groups was most often perceived as negatively affecting PrEP use. AGYW described difficulties explaining PrEP to their mothers, who believed PrEP was HIV treatment or would lead to HIV

information. To preserve participant confidentiality, the University of Cape Town's Human Research Ethics Committee has restricted these data from being shared publicly. Data access requests may be sent to the Ethics Committee at hrec-enquiries@uct.ac.za. Access may be granted to those wishing to utilize data from this study for research purposes.

**Funding:** This research was supported by grants R01MH107251 (PIs: Bekker and Celum) and F31MH119965 (PI: Giovenco) of the National Institutes of Health. The funders had no role in study design, data collection and analysis, decision to publish, or preparation of the manuscript.

**Competing interests:** The authors have declared that no competing interests exist.

infection. Disclosure to household members was notably meaningful for AGYW (both positively and negatively). Respondents reported leveraging supportive relationships for pill reminders. For respondents who perceived a household member would be unsupportive, however, non-disclosure was less feasible and PrEP use was often stigmatized. To avoid stigma, several respondents hid or discontinued PrEP.

## Conclusions

While supportive relationships may facilitate PrEP use, disclosure can also lead to stigma. Counselors should support AGYW in disclosing to key people in their social networks and provide AGYW with materials that lend credibility to explanations of PrEP. Community education is necessary to alleviate PrEP-related stigma and facilitate disclosure.

## Introduction

Nearly 25% of all new HIV infections in sub-Saharan Africa occur among adolescent girls and young women (AGYW) aged 15–24 years [1]. In South Africa specifically, estimated HIV incidence among AGYW is four times that of their male counterparts [2]. Oral anti-retroviral pre-exposure prophylaxis, or PrEP, is an efficacious biomedical HIV prevention strategy with enormous potential to reduce HIV acquisition in key populations globally [3]. Adherence to PrEP, however, has been inconsistent among young African women [4–6]. Further, adolescents and young people, and particularly adolescents under 18 years of age, remain inadequately represented in biomedical HIV prevention research posing a significant challenge for integrating PrEP into effective prevention programs and identifying gaps in adherence [7].

In qualitative data from the VOICE trial, women (19–40 years) identified a lack of support from family members, peers, partners, and the community as a contributor to poor adherence [8]. Disclosure of PrEP use can be a means of eliciting social support and may be an especially important factor to consider with respect to adolescents and young people. Adolescent development takes place within a complex web of family, peer, community, societal, and cultural influences [9]. Since most adolescents still live at home, parents can have a strong influence on adolescent motivations, decisions, and behaviors relating to health [10]. Older adolescence and young adulthood often mark a developmental transition to strong peer and partner relationships. While the identification of supportive relationships can be leveraged to promote sustained behavior change during this period of rapid life transition [11], there is a lack of research exploring the perceived role of diverse social influences on PrEP use among AGYW.

Given the disparities in HIV incidence faced by AGYW in generalized HIV epidemic settings, the relative sparsity of adolescents in PrEP research, and low adherence as an emerging barrier to PrEP effectiveness in this age group, research is needed to inform PrEP implementation efforts among AGYW with a focus on maximizing adherence. We analyzed qualitative data from the 3Ps for Prevention study [12] to explore South African AGYW's perceptions about the role of disclosure and social support in their adherence to oral PrEP. The aims of this research were firstly, to describe experiences of AGYW disclosing their PrEP use and secondly, to better understand the perceived effect of disclosure on PrEP use behavior.

## Methods

### Study design, recruitment, and enrollment

The 3Ps for Prevention study, or 3P, was a pilot prospective intervention study evaluating PrEP acceptability and adherence among HIV-uninfected AGYW taking open-label daily oral PrEP [12]. 3P was conducted at a youth-friendly clinic in a peri-urban settlement located outside Cape Town, South Africa from 2017–18. 200 HIV-uninfected AGYW were recruited through household recruitment or a social marketing campaign about PrEP. AGYW were eligible if they were HIV-uninfected, aged 16–25 years, sexually active, and intending to take PrEP and be followed for 12 months. PrEP was provided in conjunction with a standard of care HIV prevention package, which included HIV testing and counseling, condoms, and testing for sexually transmitted infections. At enrollment, AGYW were randomized to receive (or not receive) a small incentive conditional on high adherence on the basis of tenofovir diphosphate (TFV-DP) concentrations in dried blood spots (DBS). Additional detail on the study is described elsewhere [12].

### Qualitative subset with serial in-depth interviews

To better understand decisions to initiate PrEP and barriers and facilitators to daily oral PrEP use over time, a subset of 3P participants were purposively selected to participate in serial in-depth-interviews (IDIs) at months 1, 3, and 12 of study follow-up. Participants were selected to represent a range of ages and both study arms (incentive arm and standard of care).

### Data collection and processing

Socio-demographic data were collected from 3P participants at baseline and during a formative, social marketing campaign phase of the study. Participants completed a behavioral survey that assessed partnership characteristics, PrEP use disclosure, and social support at enrollment and all follow-up visits occurring at months 1, 2, 3, 6, 9, and 12. Additionally, to assess cumulative adherence over the prior two months, DBS were collected at months 1, 2, 3, 6, and 12.

Qualitative interviews followed semi-structured guides aimed at exploring: 1) AGYW decision-making and motivations surrounding PrEP initiation and continuation; 2) how AGYW incorporated daily pill-taking into their daily routines, including the types of cues and social support they used to remind themselves to take PrEP and where they stored their pills; and 3) whether AGYW disclosed their PrEP use to others, how those individuals responded, and the influence of PrEP use on other prevention behaviors, self-efficacy, and relationship dynamics (see S1 File for disclosure and relationship questions). Interviews were conducted by a local isiXhosa-speaking social scientist experienced in qualitative interviewing. Mock interviews were conducted for interviewer training and to pre-test the interview guide. Respondents could choose to have the interview conducted in isiXhosa or English, as well as the location of the interview (at their residence or the research site). Interviews lasted approximately 1–1.5 hours. All sessions were audio recorded, transcribed verbatim in English by a staff member proficient in both isiXhosa and English, and reviewed for quality control by a second staff member.

### Analysis

We conducted descriptive analysis of qualitative respondent socio-demographic data. We then described total self-reported disclosures to partners, parents, siblings, other family members, friends, and neighbors among qualitative respondents at enrollment and each follow-up visit, as well as the proportion of supportive (versus unsupportive) reactions following disclosure

experiences. Next, to contextualize qualitative findings, we described levels of PrEP adherence through months 1, 3, and 12 (when the IDIs were conducted) using TFV-DP concentrations in DBS. Detectable PrEP was defined as TFV-DP >16.6 fmol/punch (i.e., above the lower limit of quantification) at a given time point. High adherence was defined as TFV-DP ≥500 fmol/punch at month 1 prior to reaching steady state drug levels and ≥700 fmol/punch at all subsequent follow-up time points. Thresholds for high adherence were based on 100% PrEP efficacy among men who have sex with men [13,14]. Respondents with a missing DBS result who were known to have discontinued PrEP based on study records were considered to be non-adherent.

For qualitative analysis, a codebook was iteratively developed to understand AGYW's perceptions of their needs and decision-making processes from initial PrEP awareness through PrEP initiation, adherence, and persistence [15], as well as discussions surrounding disclosure or a lack of disclosure among AGYW. Transcripts were coded by three members of the study team using a shared dataset in Dedoose software (version 8.2.18, Los Angeles, CA). To assess for inter-coder reliability, the coders independently reviewed a subset of transcripts and then examined the degree to which the coding scheme was consistently applied (≥80% reliability). Another member of the research team helped resolve inconsistencies across coders and minor changes were made to the codebook to capture emergent themes. Once data were coded, excerpts under specific codes and sub-codes were compiled across all interviews and data were categorized under specific social groups that respondents reported disclosing to. Data for each social group were then described in terms of 1) experiences disclosing PrEP use and 2) the perceived effect of disclosure and social support on PrEP use throughout study follow-up.

## Ethical considerations

All study procedures were approved by the University of Cape Town's Human Research Ethics Committee (Ref: 567/2016). Participants provided written informed consent in English or isiXhosa. A parental consent waiver was granted for participants ages 16–17 years. Qualitative respondents received 200 ZAR (approximately 15 USD) for participating in each interview.

## Results

### Respondent characteristics

Table 1 lists sociodemographic and behavioral characteristics of the 22 qualitative respondents at baseline. Respondents had a median age of 20.5 years. Nearly all (n = 21; 95%) had a primary sexual partner. Among those, eight (36%) did not know their partner's HIV status. Nearly two-thirds of qualitative respondents (n = 14; 64%) were "worried a lot" about getting HIV.

In baseline and follow-up quantitative surveys, qualitative respondents most frequently self-reported disclosing to friends, with a total of 36 disclosures to friends reported throughout study follow-up (Table 2). Neighbors were the least commonly disclosed to, with 12 total disclosures reported. Further, respondents reported a total of 19 disclosures to parents and 24 disclosures each to partners, siblings, and other family members throughout study follow-up. Respondents reported they received supportive reactions following disclosure for the majority of the disclosures reported during follow-up (≥96%), with slightly fewer supportive reactions from parents (84%), neighbors (92%), and friends (94%). At month 12, almost half of respondents (47%) had detectable levels of TFV-DP and 11% were highly adherent to PrEP, based on TFV-DP ≥500 fmol/punch at month 1 and ≥700 fmol/punch at month 2 and beyond (Table 3).

**Table 1. Baseline qualitative sample characteristics.**

|  | N (%) or median (range) |
| --- | --- |
| Total Respondents | 22 |
| Completed 3 interviews | 18 (82%) |
| Completed 2 interviews | 2 (9%) |
| Completed 1 interview | 2 (9%) |
| Age (median, range) | 20.5 (16–25) |
| Age |  |
| 16–17 | 5 (23%) |
| 18–19 | 5 (23%) |
| 20–22 | 7 (32%) |
| 23–25 | 5 (23%) |
| Completed secondary school or higher[1] | 9 (41%) |
| Sometimes/often worried about food in past 30 days[1] | 17 (77%) |
| Has a primary sexual partner | 21 (95%) |
| HIV status of primary sexual partner |  |
| HIV positive | 0 (0%) |
| HIV negative | 14 (64%) |
| Respondent doesn't know | 8 (36%) |
| "How worried are you about getting HIV?" |  |
| Not worried | 2 (9%) |
| Worried some | 6 (27%) |
| Worried a lot | 14 (64%) |
| Randomized to incentive arm | 11 (50%) |

[1]Questions asked at study recruitment only.

**Table 2. Disclosure and social support report since last study visit.**

| Study visit month (Total non-missing observations) | M0 (n = 22) | M1 (n = 22) | M2 (n = 20) | M3 (n = 20) | M6 (n = 17) | M9 (n = 15) | M12 (n = 14) | Follow-up totals[3] |
| --- | --- | --- | --- | --- | --- | --- | --- | --- |
| **Disclosed to a partner, n (%)[1]** | 12 (55%) | 10 (45%) | 6 (30%) | 7 (35%) | 1 (6%) | 0 (0%) | 0 (0%) | 24 |
| Supportive reaction, n (%)[2] | 12 (100%) | 9 (90%) | 6 (100%) | 7 (100%) | 1 (100%) | -- | -- | 23 (96%) |
| **Disclosed to a parent, n (%)** | 14 (64%) | 9 (41%) | 4 (20%) | 6 (30%) | 0 (0%) | 0 (0%) | 0 (0%) | 19 |
| Supportive reaction, n (%) | 14 (100%) | 9 (100%) | 3 (75%) | 4 (67%) | -- | -- | -- | 16 (84%) |
| **Disclosed to a sibling, n (%)** | 11 (50%) | 10 (45%) | 4 (20%) | 7 (35%) | 2 (12%) | 1 (7%) | 0 (0%) | 24 |
| Supportive reaction, n (%) | 11 (100%) | 10 (100%) | 4 (100%) | 7 (100%) | 2 (100%) | 1 (100%) | -- | 24 (100%) |
| **Disclosed to other family, n (%)** | 12 (55%) | 11 (50%) | 6 (30%) | 7 (35%) | 0 (0%) | 0 (0%) | 0 (0%) | 24 |
| Supportive reaction, n (%) | 12 (100%) | 11 (100%) | 6 (100%) | 7 (100%) | -- | -- | -- | 24 (100%) |
| **Disclosed to a friend, n (%)** | 15 (68%) | 13 (59%) | 7 (35%) | 10 (50%) | 3 (18%) | 3 (20%) | 0 (0%) | 36 |
| Supportive reaction, n (%) | 12 (80%) | 13 (100%) | 7 (100%) | 8 (80%) | 3 (100%) | 3 (100%) | -- | 34 (94%) |
| **Disclosed to a neighbor, n (%)** | 1 (5%) | 7 (32%) | 2 (10%) | 2 (10%) | 1 (6%) | 0 (0%) | 0 (0%) | 12 |
| Supportive reaction, n (%) | 1 (100%) | 7 (100%) | 2 (100%) | 1 (50%) | 1 (100%) | -- | -- | 11 (92%) |

Disclosure at enrollment (M0) = disclosure of "plans to use PrEP".

Disclosure during follow-up (M1-M12) = disclosure of actual PrEP use.

[1]% = number who disclosed to at least one member of each group/total non-missing observations.

[2]% = number reporting supportive reaction following disclosure/number who disclosed to each group.

[3] Follow-up totals = sum total of disclosures reported during M1-M12 and the proportion of total disclosures that were supportive, n (%).

**Table 3. PrEP adherence for qualitative sample (n = 22).**

| Adherence | M1 (n = 21) | M3 (n = 21) | M12 (n = 19) |
|---|---|---|---|
| **Detectable PrEP, N (%)** | 21 (100%) | 20 (95%) | 9 (47%) |
| **High adherence, N (%)** | 14 (67%) | 10 (48%) | 2 (11%) |

Detectable PrEP was defined as TFV-DP >16.6 fmol/punch and high adherence was defined as TFV-DP $\geq$500 fmol/punch (month 1) and $\geq$700 fmol/punch (months 3 and 12) at a given time point.

## Qualitative findings

A total of 22 AGYW completed at least one IDI. Eighteen qualitative respondents completed all three serial interviews at approximately months 1, 3, and 12 of follow-up (Table 1). Three qualitative respondents left the study prior to completing all interviews (one respondent each following their month 1, 2, and 3 visits) and one respondent missed their month 3 interview. Qualitative findings were categorized by the social groups with whom respondents discussed disclosing, including partners, mothers, fathers, siblings, other family members (i.e., cousins, aunts, uncles, grandparents), friends, and neighbors. We described disclosure experiences and the perceived effect of disclosure and social support on PrEP use separately for each social group. Quotes are presented with respondent age at enrollment and study visit month.

**Partners.** Disclosure to partners often initially raised concerns that respondents would be unfaithful or have multiple partners because they would no longer be concerned about acquiring HIV while taking PrEP. Respondents described fears of contracting HIV through another source, such as helping an injured person or being raped, to their partners to alleviate trust issues. For example, one respondent described how she explained PrEP to her partner:

> *"[It's] Not that I'm saying that you are going to infect me or whatsoever, but I'm just doing it because we don't know what might happen. Maybe I might help an injured person who is infected and then get infected, or maybe I can go out partying and then have a blackout, have that guy take advantage of me or whatever"* (17yrs, M3).

Once partners understood these explanations, many were described as being accepting of PrEP. However, one respondent described an experience of violence following disclosure: *"When I came here for the first time and I got my first bottle of medication and I took it to him and show him and it created a problem as if I do not trust him. After he forced himself to me and he overpowered me and we ended up having sex without a condom"* (22yrs, M3).

Positive impacts of disclosure to partners included encouraging discussions surrounding HIV and testing within their relationships and receiving support from partners in the form of pill reminders: *"He was sending me messages via WhatsApp whenever he remembers and asks me if I have taken my pill"* (17yrs, M12). Many partners also expressed interest in PrEP for themselves. One respondent explained: *"I told my boyfriend and he was reminding me and sometimes if we are together and it's time for my pill, he wanted to take the pill himself and I would tell him no these are only for me not for sharing"* (19yrs, M12).

Two partners directly requested AGYW discontinue PrEP due to fears of promiscuity:

> *"He had this idea that if I took the pills, I would have multiple partners because I would be safe from getting HIV. It was more or less trust issues. . . he kept on pushing me to stop taking the pills, but I wouldn't. . . He was against it for like four months and then afterwards he saw that I was determined to take the pills, so he was like, 'Okay fine, do whatever you wanna do'"* (19yrs, M12).

**Mothers.**   Respondents described difficulties explaining PrEP to their mothers due to normative beliefs that pills are for those who are sick or that PrEP is HIV treatment or would lead to their daughter becoming infected with HIV. One respondent explained: *"My mother will always tell me that she doesn't like what we are doing, because she feels that it's inviting sickness; we are taking pills every day like we are sick" (17yrs, M3)*. Another respondent described an experience of stigma following disclosure to her mother: *"Like I am taking my pills at 21:30, so if there is someone in the house, she will tell that person I am taking pills for AIDS" (19yrs, M1)*.

Study materials (brochures, posters) and outside sources (TV or radio ads) were helpful in legitimizing explanations of PrEP to mothers. One respondent explained, *"My mom didn't believe me at first- she thought I was HIV positive- I explained to her and also showed her the posters and then she understood" (24yrs, M1)*. Another respondent described how seeing PrEP on TV helped her mother understand: "She understands it now. *They were talking about PrEP on TV the one time. . .at least now she believes it" (22yrs, M3)*.

Many respondents described a lack of support and stigmatizing comments from mothers that affected their PrEP use. To avoid stigma, respondents described implementing covert-use strategies. For example, one respondent explained: *"I hid them. I wasn't taking them in front of her, she thought I had thrown them away. . ." (19yrs, M12)*. Two respondents specifically described discontinuing their PrEP use to appease their mothers: *"[I] decided that in order to please her let me stop the pill. . . I was so mad at myself that I was no longer in charge of my body. . . if my mother had supported me I would not have stopped taking PrEP. . ." (17yrs, M12)*.

Many respondents who continued PrEP described the support they received from their mothers as evolving over time. Several mothers were described as playing an integral support role, even encouraging respondents with pill reminders. One respondent explained: *"The reason why I take [PrEP] every day, it's because my mother goes there, she reminds me" (20yrs, M1)*.

**Fathers.**   Disclosure to fathers was less common than mothers and typically discussed when the father was a respondent's primary caregiver. Most respondents did not describe difficulties disclosing to their fathers and responses were typically supportive, *"I explained to my father, reminding him of the pamphlet he saw, and I told him that I joined that study, so I am taking the pill. He said, 'That is great my daughter'" (17yrs, M12)*. Another respondent explained, *"My dad has [HIV]. . . That's why he pushed me; he's supporting me to do all this" (21yrs, M1)*.

Few respondents described reasons for not disclosing to their fathers: *"Because of my father's condition, always drunk, whether I tell them or not it's going to be the same so am not going to bother myself" (22yrs, M1)*.

Supportive fathers often encouraged respondents with pill reminders: *"My father [was supporting me] . . . He was reminding me constantly that I must take my PrEP" (21yrs, M12)*. Nevertheless, PrEP-related stigma was described by some respondents following disclosure, particularly related to beliefs about PrEP promoting promiscuity. As one respondent explained, *"Sometimes [I felt that people were judging me], because my father sometimes used to say when you are taking PrEP, for him it means that you are sleeping around" (24yrs, M3)*.

**Siblings.**   Disclosure to siblings was most often described for siblings who lived in the household: *"They only see me at home taking pills. . . Like my sister [who] is HIV positive [said], 'Why do those pills make such a noise like mine'? No, don't worry man, I'm preventing here" (17yrs, M1)*.

Siblings were often described as being more understanding of respondent explanations of PrEP as compared to parents, which helped ease the disclosure process. One respondent explained that her sister said she would support her PrEP use, but she should not be taking more risks because she was on PrEP: *"My sister told me she will support me, but I should not be doing things I shouldn't do because I am on PrEP" (19yrs, M3)*.

Siblings, and particularly siblings living in the same household, encouraged respondents with pill reminders: *"[My brother] made sure that I have water to drink the tablet, even if I sleep and don't hear the alarm, he would say wake up it's time to take the tablet, remember, then I would take it" (22yrs, M1).* Several respondents had siblings who were also taking PrEP as part of the study, which helped facilitate adherence. One respondent explained: *"I am staying with my sister and she was also on PrEP so we used to remind each other" (18yrs, M12).*

Lastly, another respondent who disclosed to a sibling outside the household explained: *"[My sisters] were not supportive because I was not staying with them" (24yrs, M12).*

**Other family members (cousins, aunts, uncles, and grandparents).**   Other family members were described as being supportive and "proud" of respondents for being proactive about their health, but were not always disclosed to, especially those living outside of the respondent's household or community. One respondent described:

> *"They were like very proud as if I have done something big and the way my granny was looking at me when I told her, like this child is taking very good care of her health. She was very proud and I was also feeling very proud and like I am a boss now" (21yrs, M3).*

Another respondent explained, *"One day my uncle. . .saw me taking my pills and asked if I am sick, I told him that I am not sick these pills prevent HIV. My uncle was so supportive he even suggested that his daughter must also join the study" (24yrs, M12).*

Disclosure to aunts, uncles, and cousins was most frequently discussed, with disclosure to grandparent less commonly reported. Similar to parents, study materials and outside sources helped legitimize explanations of PrEP to family members who were not familiar with PrEP: *"My aunt was also skeptical. . .but she started believing when she heard it on the radio" (22yrs, M1).*

Family members who were caregivers for respondents were more often disclosed to and were thus described as playing a larger support role, but some respondents described receiving support from "remote" family members in the form of pill reminders: *"[My cousin] used to ask me if I have taken the pill for that day each time I am on WhatsApp and I would tell her that I have my dear sister and she will remind me not to forget that I am a pill person. . ." (17yrs, M12).*

**Friends.**   Reactions from friends were mixed–many friends did not believe that the respondent was taking PrEP (and not ART) and that PrEP was safe:

> *"I told [my friends] about PrEP and we argued—they said there is no such thing you will all be sick from what you are taking. . . until they saw a post from Facebook. One of my friends came and told me that she saw the post so it was true what I said and she was interested to join the study as well. . ." (22yrs, M12).*

Further, another respondent described: *"One of my friends said she had heard that the pills we are taking are ARVs . . . To prove the point, I told her to go to the chemist and ask if they have Truvada. . .she must ask what Truvada is for" (20yrs, M3).* Several respondents who perceived they would be stigmatized for taking PrEP by friends chose not to disclose their PrEP use.

Stigma and a lack of support from friends were not described as negatively affecting PrEP use as described for other social groups. Supportive friends, however, encouraged respondents with pill reminders: *"They would remind me to pack my pills if we are going out because we might not come back" (20yrs, M12).* Several respondents had friends who were also in the study, which was consistently described as a source of support: *". . .I had one friend that I was really close to and she also was taking PrEP, so it was easy for me to talk about things, and I would also remember to take my pills when I saw her taking hers. . ." (19yrs, M12).*

**Neighbors.**   Disclosure to neighbors was least commonly described by respondents. One respondent who disclosed to a neighbor living with HIV explained: "*. . .my neighbor has this HIV virus. . . He [was] saying PrEP is good and if PrEP was introduced before he got this virus maybe, not maybe, for sure, he [would have] participated in the study" (22yrs, M1).* A few respondents described unintended disclosure to neighbors that exposed them to stigma:

*"One day someone I grew up with saw [my pills] next to the TV and shook them and they gave the* same *noise as the ARVs. . .I explained [PrEP] to her. . . she said [to] my face there is no such thing. . . She spread those rumors to others as well" (24yrs, M12).*

Stigma from neighbors caused several respondents to hide their PrEP pills and was a source of stress for respondents. One respondent explained:

*"[I told] my neighbor about PrEP. . . She said that these pills mess up someone's shape–if you are fat you get thin and if you are thin you get fat. I do not like to put them where people can see them because they will ask questions and tell me things. . ." (24yrs, M3).*

Another respondent described: *"There was a challenge at first, some of the people that I am staying with in my yard, they thought that I am on ARVs. . .I was stressed but I told myself that it is because they do not have the knowledge" (25yrs, M12).*

## Discussion

Serial in-depth interviews among 22 South African AGYW in an open-label PrEP demonstration project indicate that disclosure is an important factor that can be a facilitator or barrier to their oral PrEP use. AGYW frequently reported disclosing their PrEP use to partners, parents, siblings, other family members, and friends. While the majority of disclosures (≥84%) were described as supportive in a quantitative survey, IDI data revealed more variability in responses. AGYW described disclosure experiences that enabled instrumental PrEP adherence support (e.g., pill reminders) from key persons in their social networks. Unsupportive disclosures, however, could undermine their adherence to PrEP and subject AGYW to stigma.

Disclosure to partners varied across AGYW, but descriptions were largely similar to African women enrolled into the FEM-PrEP [16]. Several AGYW chose not to disclose their PrEP use to partners and thus did not include their partners in discussions surrounding their PrEP support networks. This was particularly common among younger AGYW who did not live with their partners or were in casual partnerships and perceived their partner would not approve. Among AGYW who did disclose, partners were often described as accepting or supportive of respondent PrEP use. For a few AGYW, however, disclosure of PrEP use altered perceptions of trust in their relationships, resulting in their partners requesting they discontinue PrEP and, in one case, intimate partner violence. In the VOICE-C study, African women and their male partners also described a disruption in their relationship power dynamic as a result of PrEP [17]. VOICE-C partner narratives suggest that partner education on PrEP and allowing partners to speak with clinic staff can help ease the disclosure process.

Disclosure to mothers was most frequently discussed in IDIs, and a lack of support from this group was most often described as negatively affecting adherence. Disclosure to mothers was often complex. Respondents had a hard time explaining PrEP to their mothers, who didn't view their daughters as credible sources and believed that PrEP was HIV treatment or would lead to HIV infection. African women in HPTN 067/ADAPT also described experiences of community distrust in PrEP and women's participation in PrEP research [18], which can lead to the development of stigmatizing attitudes. Since many younger AGYW in 3P lived with

their mothers, some implemented covert-use strategies (e.g., hid their PrEP pills or took PrEP away from the home), which were typically described as negatively affecting adherence, or discontinued PrEP to avoid stigma. Adherence programs should consider the important role parents play in most young people's lives and provide AGYW, and particularly younger AGYW who live at home, with strategies for disclosure to this important group.

Disclosure to household members, which often included parents, siblings, other family members, and partners, was consistently discussed as having a substantial effect on PrEP use behavior, both positively and negatively. This may be a unique group to consider targeting with respect to harnessing support for young people initiating PrEP, given that adolescents are guided by diverse network of social influences that change rapidly as they transition to young adulthood [9–11]. A positive effect of disclosure to household members in this study included leveraging supportive relationships for daily pill reminders. For AGYW who perceived that a household member would be unsupportive, however, non-disclosure was less feasible since it was difficult for them to conceal their pills. When unintended and/or incomplete disclosure occurred, PrEP use was often stigmatized. Therefore, adherence counseling programs should also support AGYW in disclosing (or not disclosing) to individuals in their households.

Across social groups, stigma was a persistent theme resulting from both intentional and unintentional disclosure experiences. Stigma towards PrEP, however, was often described as evolving over time, particularly when outside sources (e.g., TV or radio ads, study brochures, a local study staff member) verified explanations of PrEP. To alleviate PrEP-related stigma and facilitate disclosure, materials that reinforce PrEP's legitimacy should be provided to AGYW initiating PrEP, and education and awareness campaigns should engage trusted sources (e.g., community members, traditional healers) and venues (e.g., faith-based venues, community organizations) to educate communities more broadly [18]. In addition, research has highlighted the need to control the narrative surrounding PrEP to shift the focus away from stigma. For example, instead of perpetuating the message that PrEP is "for people at very high risk for infection," PrEP programs could emphasize that PrEP is "for people who want to reduce their anxiety about HIV infection and take greater responsibility for their sexual health" [19].

Finally, support for PrEP following disclosure in IDIs was typically described in terms of instrumental support, such as pill reminders. Other forms of support, such as emotional support or encouragement, were less frequently described as directly affecting PrEP use. However, several respondents described individuals (primarily family members) expressing that they were "proud" of respondents for taking PrEP, and friends and siblings who were also taking PrEP provided AGYW with a sense of community. Given that community clubs have been found to improve adherence to antiretroviral therapy among those living with HIV [20], these less explicit forms of support may also influence PrEP use behavior among AGYW. Further, in HPTN 067/ADAPT, "PrEP champions" emerged as a group of PrEP advocates who could be leveraged to provide peer support and even shift community beliefs about PrEP [16]. Last, while respondents did not discuss disclosure to clinic staff and healthcare providers, these groups may play an important support role in AGYW's PrEP journey outside a research setting.

There are several limitations to this research. First, given the qualitative nature of this investigation, our sample size was small, and our findings might not generalize to other populations. Moreover, this investigation relied primarily on a subjective assessment of respondents' self-reported disclosure experiences using qualitative methods. While a quantitative survey also collected self-reported disclosure and reactions at each study visit, this data is subject to recall and information bias and does not adequately capture nuances in disclosure experiences. Further, this research does not provide a quantitative assessment of the association between disclosure and PrEP use, limiting our understanding of this relationship.

This qualitative investigation informs several recommendations for future research. First, research should examine the association between PrEP-use disclosure and adherence quantitatively. Second, given that stigma was consistently described across social groups and the perceived negative effect of stigma on PrEP use and general well-being, concerted efforts should be made to develop and evaluate stigma-reducing messaging surrounding PrEP for AGYW. Third, research is needed to inform best practices for identifying support networks and leveraging support from PrEP allies to promote the formation and continuation of early adherence behaviors. Lastly, for AGYW who perceive a lack of support for daily pill-taking, continued research is needed to examine the acceptability of alternative PrEP modalities (e.g., topical, injectable, insertable) and dosing regimens (e.g., intermittent, on-demand). This is particularly needed since long-acting cabotegravir was recently found to be safe and superior to daily oral PrEP for preventing HIV infection among African women in HPTN 084 [21].

## Conclusions

While supportive relationships may help to facilitate adherence, disclosure of PrEP use can also subject AGYW to stigma and violence and should be considered on an individual basis. Careful age and developmentally appropriate counseling for AGYW initiating PrEP should support the explanation of PrEP to the important people in their lives, including parents, partners, and other individuals residing in the household or with whom unintended disclosure may be likely. Materials that lend credibility to explanations of PrEP and reinforce PrEP's legitimacy should be provided to AGYW initiating PrEP. Further, community education on the rationale and importance of PrEP is necessary to increase awareness and alleviate PrEP-related stigma.

## Supporting information

**S1 File.**
(DOCX)

## Acknowledgments

We would like to thank the 3P study team and the young women who participated in this study.

## Author Contributions

**Conceptualization:** Ariane van der Straten, Connie L. Celum, Linda-Gail Bekker.

**Data curation:** Danielle Giovenco, Shannon O'Rourke.

**Formal analysis:** Danielle Giovenco, Shannon O'Rourke.

**Investigation:** Ariane van der Straten, Connie L. Celum, Linda-Gail Bekker.

**Methodology:** Ariane van der Straten, Connie L. Celum, Linda-Gail Bekker.

**Project administration:** Katherine Gill, Lauren Fynn, Menna Duyver, Jennifer F. Morton.

**Resources:** Katherine Gill, Lauren Fynn, Menna Duyver, Shannon O'Rourke, Ariane van der Straten, Jennifer F. Morton, Connie L. Celum, Linda-Gail Bekker.

**Supervision:** Katherine Gill, Ariane van der Straten, Jennifer F. Morton, Connie L. Celum, Linda-Gail Bekker.

**Visualization:** Danielle Giovenco.

**Writing – original draft:** Danielle Giovenco.

**Writing – review & editing:** Katherine Gill, Lauren Fynn, Menna Duyver, Shannon O'Rourke, Ariane van der Straten, Jennifer F. Morton, Connie L. Celum, Linda-Gail Bekker.

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
