## [Decision Letter · Decision Letter 0]

11 Dec 2020

PONE-D-20-33614

Experiences of PrEP use disclosure among South African adolescent girls and young women and its perceived impact on adherence

PLOS ONE

Dear Dr. Giovenco,

Thank you for submitting your manuscript to PLOS ONE. Before we accept the manuscript for publication, there are a few issues we would like you to address - see comments/suggestions including in attached document.

We look forward to receiving your revised manuscript.

Kind regards,

Webster Mavhu

Academic Editor

PLOS ONE

Journal Requirements:

2. Please include additional information regarding the survey or questionnaire used in the study and ensure that you have provided sufficient details that others could replicate the analyses. For instance, if you developed a questionnaire as part of this study and it is not under a copyright more restrictive than CC-BY, please include a copy, in both the original language and English, as Supporting Information, or include a citation if it has been published previously.

3. In the Methods, please discuss whether and how the questionnaire was validated and/or pre-tested. If these did not occur, please provide the rationale for not doing so.

4.We note that you have indicated that data from this study are available upon request. PLOS only allows data to be available upon request if there are legal or ethical restrictions on sharing data publicly. For information on unacceptable data access restrictions, please see http://journals.plos.org/plosone/s/data-availability#loc-unacceptable-data-access-restrictions.

5. Please amend your list of authors on the manuscript to ensure that each author is linked to an affiliation. Authors’ affiliations should reflect the institution where the work was done (if authors moved subsequently, you can also list the new affiliation stating “current affiliation:….” as necessary).

Additional Editor Comments (if provided):

This is a well-written paper on an important topic. I would largely ignore the reviewer's comments on Discussion and limitations.

Given the recently-released amazing HPTN 084 study results, paper needs to refer even in passing - see suggestion in attached document.

See additional comments/suggestions in attached paper.

I would say qualitative respondents (not participants) throughout.

Line 77 refers to just oral PrEP, needs to be revised.

Include interview guide as supporting file and reference it as S1 file.

See reviewer's comment on where interviews took place.

Data processing section needs to say how accuracy of translation from isiXhosa to English was ensured. Reading the quotes, it looks like some of the grammar is due to inaccurate translations?

Check lines 302-3.

Results section on other family members could be more nuanced – Was there a pattern looking at male vs. female or younger vs. older?

Reviewers' comments:

Reviewer's Responses to Questions

**Comments to the Author**

1. Is the manuscript technically sound, and do the data support the conclusions?

Reviewer #1: Yes

2. Has the statistical analysis been performed appropriately and rigorously? 

Reviewer #1: N/A

3. Have the authors made all data underlying the findings in their manuscript fully available?

Reviewer #1: Yes

4. Is the manuscript presented in an intelligible fashion and written in standard English?

Reviewer #1: Yes

5. Review Comments to the Author

Reviewer #1: Data collection section- can you please add the venue where the interviews were conducted as this has a potential effect on AGYW's responses

line 221 not very clear

the discuss section reads more like a repetition of the findings section and requires improvement. the discussion should outline the implication of the findings on the promotion of PrEP uptake. What the key message to clinicians or PrEP interventions based on these finding what are the key areas to look out for, What should be the main focus of PrEp messaging going forward.

line 348 on limitations Qualitative study design findings are never meant to be generalized this is know and this cant be a limitation because that is how qualitative findings are designed. Before one conduct a qualitative result they know what it can and cannot do and this cant be a limitation just like you can not mention the generalizability of quantitative findings as a limitation. Same with the subjective nature of the qualitative studies. The limitations you have cited challenges the whole idea of qualitative research having a distinct epistemological standpoint which is different from a quantitative research.

6. PLOS authors have the option to publish the peer review history of their article (what does this mean?). If published, this will include your full peer review and any attached files.

Reviewer #1: No

---

## [Author Response · Author response to Decision Letter 0]

29 Jan 2021

EDITOR COMMENTS:

This is a well-written paper on an important topic. I would largely ignore the reviewer's comments on Discussion and limitations.

Given the recently-released amazing HPTN 084 study results, paper needs to refer even in passing - see suggestion in attached document.

Response: This has been added to the Discussion where suggested and the HPTN press release was cited in the References. Please let me know if the citation is acceptable. 

See additional comments/suggestions in attached paper.

Response: Comments throughout the paper have been incorporated.

I would say qualitative respondents (not participants) throughout.

Response: This has been changed

Line 77 refers to just oral PrEP, needs to be revised.

Response: The citation referenced in this line references a review on the effectiveness and safety of oral HIV preexposure prophylaxis. Since this paper is about adherence to oral PrEP, which is likely to be very different than other PrEP formulations, we have focused the Introduction section on oral PrEP.

Include interview guide as supporting file and reference it as S1 file.

Response: The qualitative IDI guide has been included as a supporting file. Because the guide was long and was designed to explore several aspects of PrEP use, we have only included the relevant section which included disclosure and relationship questions. 

See reviewer's comment on where interviews took place.

Response: This has been added to the “Data collection and processing” section. Participants could choose to have interviews conducted at their homes or the research site. 

Data processing section needs to say how accuracy of translation from isiXhosa to English was ensured. Reading the quotes, it looks like some of the grammar is due to inaccurate translations?

Response: All sessions were audio recorded, transcribed verbatim in English by a staff member proficient in both isiXhosa and English, and reviewed for quality control by a second staff member. This detail has been added to the methods. We acknowledge there are a few grammar issues in the quotes, possibly due to translations or the casual nature of the interviews, but we preferred to leave the quotes as unchanged as possible. 

Check lines 302-3.

Results section on other family members could be more nuanced – Was there a pattern looking at male vs. female or younger vs. older?

Response: For disclosure to family members, AGYW seemed to be most likely to disclose to those who they perceived would be supportive. Therefore, we did not find any notable differences in supportive disclosures across family member age and gender. We have, however, added that disclosure to aunts, uncles, and cousins was more commonly discussed than disclosure to grandparents. This may be because AGYW perceived this older generation would be less supportive.

REVIEWER #1 COMMENTS:

Data collection section- can you please add the venue where the interviews were conducted as this has a potential effect on AGYW's responses

Response: See response above

line 221 not very clear

Response: We apologize for the typo in this sentence. The revised sentence now reads: Lastly, another respondent who disclosed to a sibling outside the household explained: “[My sisters] were not supportive because I was not staying with them” (24yrs, M12).

the discuss section reads more like a repetition of the findings section and requires improvement. the discussion should outline the implication of the findings on the promotion of PrEP uptake. What the key message to clinicians or PrEP interventions based on these finding what are the key areas to look out for, What should be the main focus of PrEp messaging going forward.

Response: Given the qualitative nature of the data and small sample size, this analysis is intended to provide support for future investigations into this topic prior to recommendations.

line 348 on limitations Qualitative study design findings are never meant to be generalized this is know and this cant be a limitation because that is how qualitative findings are designed. Before one conduct a qualitative result they know what it can and cannot do and this cant be a limitation just like you can not mention the generalizability of quantitative findings as a limitation. Same with the subjective nature of the qualitative studies. The limitations you have cited challenges the whole idea of qualitative research having a distinct epistemological standpoint which is different from a quantitative research.

Response: We present both qualitative and descriptive quantitative data. Therefore, we feel there is no harm in mentioning this important limitation, despite its potential redundancy.

---

## [Editor Report · Decision Letter 1]

22 Feb 2021

PONE-D-20-33614R1

Experiences of pre-exposure prophylaxis (PrEP) use disclosure among South African adolescent girls and young women and its perceived impact on adherence

PLOS ONE

Dear Dr. Giovenco,

A few more edits or suggestions in attached.

We look forward to receiving your revised manuscript.

Kind regards,

Webster Mavhu

Academic Editor

PLOS ONE

---

## [Author Response · Author response to Decision Letter 1]

22 Feb 2021

Dear Dr. Webster Mavhu,

We appreciate the opportunity to provide additional revisions on our manuscript. 

No papers listed in the references have been retracted. We have replaced one citation that was previously listed as “forthcoming” with the citation for the recently published manuscript. In addition, we have replaced the citation for a conference abstract with the citation of another recently published manuscript. No other changes to the references have been made. Please let us know if there are any other specific changes to this section you would like us to make.

Only one qualitative coder has been included as a co-author (SO). Therefore, for consistency, we have not included any staff initials in the qualitative analysis section.

All in-text edits made by the editor have been accepted with the exception of line 206, where the sentence remains: “Disclosure to partners often initially raised concerns that respondents would be unfaithful or have multiple partners because they would no longer be concerned about acquiring HIV while taking PrEP” to keep the verb tense consistent throughout sentence. 

We appreciate your thorough edits. Please let us know if there are any additional edits.

Sincerely,

Danielle Giovenco

---

## [Editor Report · Decision Letter 2]

24 Feb 2021

Experiences of oral pre-exposure prophylaxis (PrEP) use disclosure among South African adolescent girls and young women and its perceived impact on adherence

PONE-D-20-33614R2

Dear Dr. Giovenco,

We’re pleased to inform you that your manuscript has been judged scientifically suitable for publication and will be formally accepted for publication once it meets all outstanding technical requirements.

Kind regards,

Webster Mavhu

Academic Editor

PLOS ONE
---

## [Editor Report · Acceptance letter]

26 Feb 2021

PONE-D-20-33614R2 

Experiences of oral pre-exposure prophylaxis (PrEP) use disclosure among South African adolescent girls and young women and its perceived impact on adherence 

Dear Dr. Giovenco:

I'm pleased to inform you that your manuscript has been deemed suitable for publication in PLOS ONE. Congratulations! Your manuscript is now with our production department. 

Kind regards, 

on behalf of

Dr. Webster Mavhu 

Academic Editor

PLOS ONE